# EzMechanism: an automated tool to propose catalytic mechanisms of enzyme reactions

Antonio J. M. Ribeiro ✉, Ioannis G. Riziotis , Jonathan D. Tyzack, Neera Borkakoti & Janet M. Thornton ✉

Over the years, hundreds of enzyme reaction mechanisms have been studied using experimental and simulation methods. This rich literature on biological catalysis is now ripe for use as the foundation of new knowledge-based approaches to investigate enzyme mechanisms. Here, we present a tool able to automatically infer mechanistic paths for a given three-dimensional active site and enzyme reaction, based on a set of catalytic rules compiled from the Mechanism and Catalytic Site Atlas, a database of enzyme mechanisms. EzMechanism (pronounced as 'Easy' Mechanism) is available to everyone through a web user interface. When studying a mechanism, EzMechanism facilitates and improves the generation of hypotheses, by making sure that relevant information is considered, as derived from the literature on both related and unrelated enzymes. We validated EzMechanism on a set of 62 enzymes and have identified paths for further improvement, including the need for additional and more generic catalytic rules.

Enzymes are proteins that accelerate the chemical reactions necessary for life. These catalytic macromolecules are abundant in all cells (representing 22% of the proteins coded in the human genome and 40% in *Escherichia coli*, for example[1]) and are widely studied. Of particular interest is understanding their reaction mechanisms: the sequence of events in the active site, such as the formation and cleavage of bonds, that explains how the substrate is transformed into the products. Enzyme mechanisms are crucial to understand enzyme function and evolution. This knowledge opens the door to a host of green chemistry and biotechnological applications, from the modulation of enzyme function through rational design[2], to the prediction of the impact of enzyme variants on disease[3] and the development of drugs targeted at the active site[4].

Learning about new enzyme mechanisms is a complex task, requiring the application of diverse types of experimental and computational methods. Enzymatic reaction rates, as well as the rate dependence on pH, temperature or chemical species such as cofactors, can be inferred from kinetic assays[5]. Potential catalytic sites identified among highly conserved residues[6] can be confirmed by mutagenesis studies[7]. Spectroscopy data, such as electron paramagnetic resonance for metals and radical species[8], or fluorescence for fluorescent intermediates[9],

may confirm or exclude the presence of certain molecular species along the reaction path. Three-dimensional (3D) structures[10] provide information about the precise location of catalytic residues, substrates and cofactors in the active site. Finally, computational chemistry methods, such as quantum mechanics and molecular mechanics (QM/MM) calculations[11,12], have been used to simulate reaction mechanisms with increasingly accurate models of the enzyme. With the help of these methods, researchers have succeeded in building a rich literature about all aspects of enzyme function, including their reaction mechanisms. We have captured some of this information in the M-CSA (Mechanism and Catalytic Site Atlas)[13], including a machine-readable representation of the catalytic steps, which forms the foundation of the present work.

EzMechanism is an automated method able to generate potential enzyme mechanisms for a given 3D structure of an active site (a related method[14], developed independently, is discussed in the Methods). Proposals are generated in a matter of a minutes up to a few hours, depending on the complexity of the active site, and should be particularly useful for researchers in the initial stages of studying an enzyme mechanism, when different hypotheses for the mechanism are being considered. EzMechanism proposals should then be tested experimentally or computationally using more sophisticated methods, such as

European Bioinformatics Institute, Wellcome Trust Genome Campus, Hinxton, Cambridge, UK. ✉e-mail: ribeiro@ebi.ac.uk; thornton@ebi.ac.uk

QM/MM calculations. The current version of EzMechanism excludes radical reactions and oxidation-reduction reactions involving metals.

In the following sections, we describe how we have created a set of rules of enzyme catalysis from M-CSA data and how the search algorithm was implemented. We give an overview of the validation we have performed (complete analysis in the Supplementary Information) and present an example test case. Details and documentation of the web user interface are given in the Supplementary Information and are also available at https://www.ebi.ac.uk/thornton-srv/m-csa/EzMechanism/documentation.

## Results

### The rules of enzyme catalysis

Enzymes use a diverse set of active sites to catalyze a large number of reactions, but the building units of these active sites are limited. Less than half of the 20 amino acids frequently play a direct role in the mechanism[6], and the number of available cofactors is equally restricted[15]. This means that many enzymes will necessarily share components of their mechanisms, even if these enzymes are not evolutionarily related or if they catalyze different chemical reactions. For related enzymes the similarities will be even higher. The first step toward our goal was to identify these recurring 'mechanistic components', which we called the 'rules of enzyme catalysis'. These rules codify chemical transformations that are possible when certain chemical groups are observed in the active site (typically, there is one rule for each catalytic step) and can be written as simple chemical reaction equations. Ideally, a complete set of catalytic rules would be able to spawn all possible enzyme mechanisms when chained together.

Figure 1 summarizes the process we followed for the creation of the catalytic rules. First, we parsed the two-dimensional (2D) curly arrow diagrams of the mechanistic steps in the database (excluding radical and redox reactions involving metals) to extract the relevant information about bond changes and intervening chemical groups (Fig. 1a). Second, a graph was built representing possible electron pathways (Fig. 1b) based on the curly arrows and the atoms they touch (up to two bonds away). Finally, by traversing this graph, we extract possible rules that are based on the literature knowledge (Fig. 1c). This process leads to the creation of two types of rule: single-step rules, which are observed in their entirety in at least one mechanistic step in the database; and mixed-step rules, which cannot be found in the database but rather are the result of combining information from different steps that share similar chemical groups (Methods). In total, 7,218 catalytic rules were obtained from 691 enzymes and 2,925 catalytic steps. Of these rules, 3,668 are single-step rules, while the remaining 3,550 arise from mixing information from more than one step (mixed-step rules).

The number of catalytic rules, as well as their specificity, is dependent on the exact algorithm followed during their creation. Before settling on the current definition, we tested other ways of building the rules, such as including only the reaction centers (atoms directly involved in bond formation and cleavage) and the immediate surrounding atoms, which leads to the creation of fewer rules. Smaller and less specific rules such as these match more enzyme active sites but lead to spurious matches, as they are not able to differentiate between a hydroxyl group from a carboxylic group, for example. On the other hand, rules that include three or more shells around the reaction centers produce more meaningful matches but, by being overly specific,

they match fewer active sites and are less useful for searching as a result. We found that rules that include atoms up to two bonds away from the reaction center are a good compromise between specificity and applicability. We further refined this definition by considering that rules should not discriminate between carbon or hydrogen atoms that

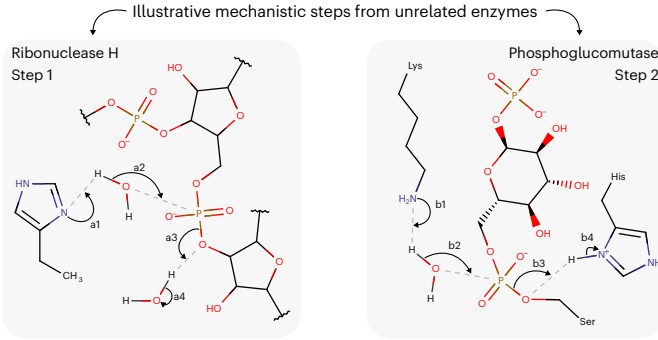

**a** Parse mechanistic steps data in M-CSA

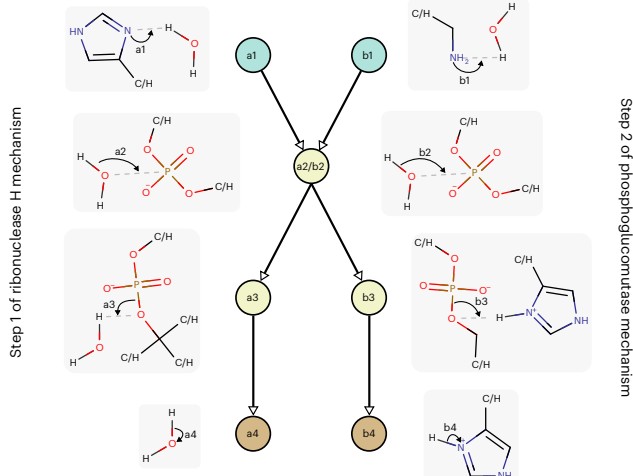

**b** Build graph of curly arrows

**c** Traverse graph and create catalytic rules

**Fig. 1 | The process followed to generate the rules of enzyme catalysis as extracted from the M-CSA database, explained for two mechanistic steps. a**, Example of two mechanistic steps from unrelated enzymes that share some chemical similarities. **b**, Graph of curly arrows showing the electron flow in the chemical groups seen in the exemplified steps. **c**, Single- and mixed-step rules that can be created by traversing the curly arrow graph. The small version of the curly arrow graph beside each rule indicates how the graph must be traversed to generate that rule.

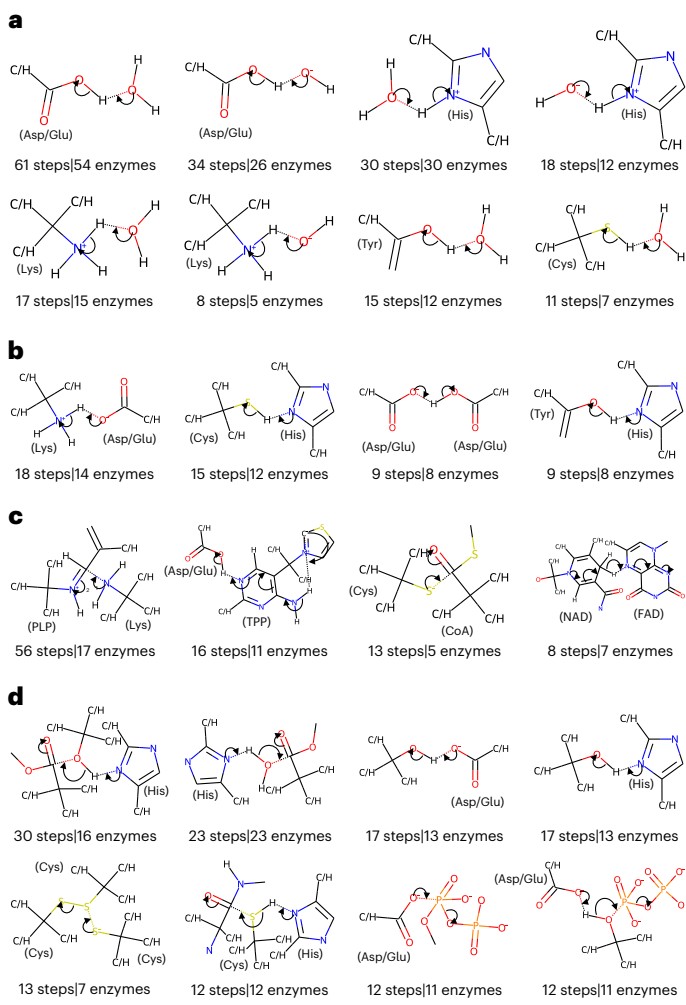

**Fig. 2 | The catalytic rules most commonly observed in the M-CSA database. a**–**d**, Rules match chemical groups, not amino acids or other molecules but, to facilitate comprehension, some molecules and amino acids that match these chemical groups are indicated in parentheses. **a**, Proton transfers with water species. **b**, Proton transfers between amino acids. **c**, Rules involving cofactors. **d**, Other common rules. Glu, glutamate; Asp, aspartate; His, histidine; Lys, lysine; Tyr, tyrosine; Cys, cysteine; PLP, pyridoxal 5′-phosphate; TPP, thiamine(1+) diphosphate; CoA, acetyl-coenzyme A; NAD, nicotinamide adenine dinucleotide and FAD, flavin adenine dinucleotide.

are two bonds way from the reaction centers. This means that formic acid and acetic acid, for example, are equivalent for rule-matching purposes, when the reaction center is the negatively charged oxygen. In Figs. 1 and 2, positions that can match both C and H are labeled 'C/H'.

The generated rules can be understood as a list of chemical groups that, if present in the active site, can be transformed in a certain way. For example, the first rule shown in Fig. 2b means that if there is a protonated amine group and a deprotonated carboxylic group in the active site, then a proton transfer can occur between the two groups. Rules do not contain any metadata about the location of these chemical groups, which means they can match any molecules regardless of their role in the mechanism (such as catalytic residues, substrates or cofactors), although in this example they will match mostly lysine and aspartate/glutamate residues. Rules are stored in the database as reaction SMARTS expressions (SMARTS is a type of line notation based on SMILES used to specificy substructural patterns in molecules) plus an additional column with information about the curly arrows representing the movement of electrons, which allow for the reconstruction of

the rule in a pictorial form. In this way, rules are both machine readable and interpretable by humans.

Figure 2 shows the most common catalytic rules identified in M-CSA. Although most rules have a corresponding reverse rule (where the reactants and products are reversed), these are not shown in the picture for simplicity. The full set of rules can be browsed at www.ebi.ac.uk/thornton-srv/m-csa/rules/, which also shows the catalytic steps from which each individual rule was extracted. In agreement with our previous analysis on the roles of catalytic residues[6], the most common catalytic rules are involved with proton transfers. The most common one, observed in 61 mechanistic steps and 54 enzymes throughout the database, is the proton transfer between a carboxylic group and a water and/or hydronium molecule (Fig. 2a). This is a common reaction step that represents the protonation and/or deprotonation of Asp and Glu amino acids by bulk water, which is often necessary to recycle the active site. A similar rule corresponds to the proton transfer between a carboxylic group and water and/or hydroxide. In fact, many common rules involve proton transfers between water molecules (to generate hydroxide, water or hydronium) and chemical groups matching specific amino acids, such as the imidazole ring (histidine), methylamine (lysine), propan-2-ol (tyrosine) or thiol group (cysteine). Proton transfers between chemical groups that match pairs of amino acids are also common (Fig. 2b): Lys and Asp and/or Glu; Cys and His; Asp and/or Glu to another Asp and/or Glu or Tyr to His. Although the third and fourth rules in Fig. 2d look like proton transfers between Ser to Asp and/or Glu and Ser to His, respectively, they come mostly from other molecule's hydroxyl groups, rather than serine residues.

The second most common rule, present in 56 steps and 18 mechanisms, represents the attack of an amine group on a pyridoxal 5′-phosphate cofactor (Fig. 2c). Indeed, cofactors tend to perform the same function across enzymes[16], so other common rules contain groups found in cofactors, such as thiamine diphosphate and coenzyme A. However, it is not always the case that the same cofactor will follow the same catalytic rule in different enzymes. NAD(P), for example, is a common cofactor, present in more than 30 steps, but since it donates and/or receives a hydride from different chemical groups, it is present in many different rules. The most common one, seen in eight steps, is a hydride transfer between NAD(P) and flavin adenine dinucleotide.

The rules discussed up to now are all single-step rules. Mixed-step rules expand the applicability of EzMechanism by mixing information from different mechanistic steps. The rationale for this is that while interacting curly arrows (representing close chemical groups or molecules) are specific to each other, this might not be the case for chemical groups or molecules that are far away in a curly arrow chain. The bottom part of Fig. 1c shows four examples of mixed-step rules with indication of which curly arrows come from the two different steps from which they were extracted.

We hope it becomes apparent from the present discussion that catalytic rules, as formulated here, can be a powerful tool to understand how enzymes operate. Furthermore, the modular approach we took for decomposing and assembling the rules mimics the evolutionary process in enzymes, in that mutations of single residues will affect only parts of the curly arrow chains[17]. A more detailed analysis of these rules and their usefulness to understand enzyme evolution is in progress.

## Automatic proposal of enzyme mechanisms

To search for the mechanism of specific enzymes, EzMechanism requires the user to supply information about the 3D structure of the active sites and about the overall chemical reaction catalyzed. The web user interface facilitates the input process by decomposing it into four steps: (1) choose a Protein Data Bank (PDB) structure; (2) choose the catalytic residues in that structure; (3) define the substrates and cofactors, and map them to ligands in the structure and (4) finally, define the overall reaction by defining their reactants and products in

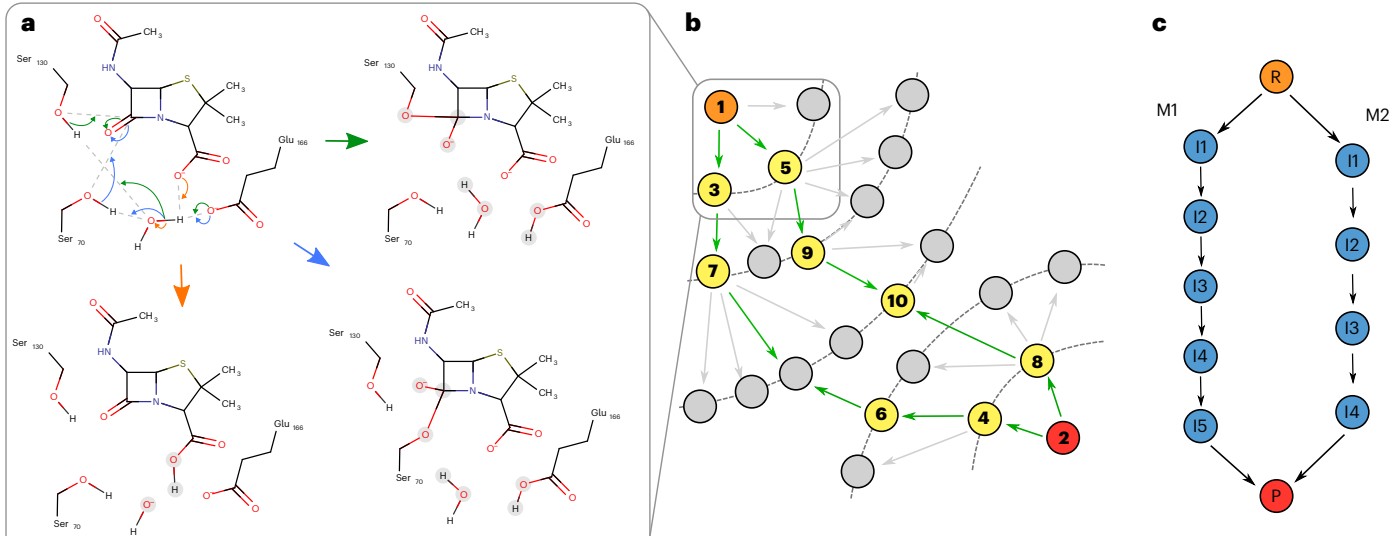

**Fig. 3 | Illustration of the mechanism search algorithm used by EzMechanism.**
**a**, Example of a configuration that matches three rules (identified by the three different colors). Each rule match leads to the creation of a new configuration. **b**, Example of a configuration graph that represents the catalytic space explored by EzMechanism. Green arrow edges represent steps in the search that are prioritized according to the prioritization algorithm. Numbers in the configurations exemplify the order followed by the software to match the rules to each configuration. Orange, reactant configuration; red, products configuration; yellow, configuration matched with rules and gray, other configurations. **c**, A cleaner representation of the two possible mechanisms that are present in **b**. Every path that links the reactants with the products configuration represents a potential mechanism.

the context of the active site. These steps are explained in detail in the Supplementary Information and the web documentation.

By combining these inputs with the catalytic rules, EzMechanism can generate a landscape of possible reaction steps around the reactants and products configurations. Ideally, in this landscape there will be one or more paths of reaction steps that link the reactants to the products, which represent possible mechanisms for the enzyme under study. Figure 3 depicts this overall process. The search algorithm starts by checking every catalytic rule against the configuration of the reactants (orange circle in Fig. 3a), and for each matching rule (representing a reaction step), a new configuration is created, representing an intermediate. The process of rule matching is similarly performed for the products configuration (red circle in Fig. 3) and then in an iterative way for the newly generated configurations (annotated in yellow). A graph is used to store the output of this search (Fig. 3b), where nodes are configurations of the active site (reactants, products and intermediates) and edges are the reaction steps (generated from rule matches) that transform these configurations into each other.

Since for a typical active site, the number of possible reaction steps generated in this way is too large to be explored exhaustively, the search is prioritized toward the most promising configurations. Gray circles in Fig. 3b represent configurations that are not checked against the rules. The gray circle inside the rounded square, for example, was generated from a rule matching the reactants configuration, but since that step was unfavorable (indicated by the gray arrow), that part of the chemical space was not explored further. Details about the prioritization function and the search algorithm are given in the Methods.

In the toy example shown in Fig. 3b there are two paths of configurations (representing a sequence of catalytic steps) that link the reactants to the products. These two possible reaction mechanisms are represented schematically in Fig. 3c.

## Validation using enzymes annotated in M-CSA

To test the applicability of EzMechanism, we started by identifying suitable candidates among the first 100 enzymes annotated in the M-CSA. We used a previously compiled dataset[18] to identify which PDB structures of each enzyme contained ligands similar to the biological substrates. For 35 of these enzymes, it was not possible to build a model containing all the catalytic residues, substrates and cofactors (only 26% of enzyme structures in the PDB have a ligand with at least 70% similarity to the cognate ligand[18]). EzMechanism could still in principle work for these enzymes after modeling of the enzyme–ligand complex. Another nine entries were excluded from the analysis because they involved either a radical mechanism (eight enzymes) or a stereoisomerase reaction (one enzyme), which are out of scope for the current version of the software. This validation, which we divided into three parts that test different parts of the software, focuses on the remaining 56 enzymes. Full results are available interactively at www. ebi.ac.uk/thornton-srv/m-csa/EzMechanism/, and are summarized in the Supplementary Information.

We first tested whether the correct mechanism (as annotated in the database) could be found for each enzyme when using rules generated solely from that same database entry. This test makes sure that the rule creation process, the matching of the rules and the overall mechanistic search are implemented correctly. EzMechanism passes this test for 55 of the 56 enzymes, failing in an unusual case where two concurrent rules of the same step (M-CSA:90 step 2) share one atom and, for this reason, cannot be matched sequentially.

Next, to test how well the prioritization algorithm works, the entire rule dataset is used for each prediction. A large number of configurations can be created (up to 100,000 in some enzymes) when matching every rule to some of these active sites and so, if the prioritization heuristics are inadequate, the correct mechanism may not be found among all the other possibilities (the software does not rank the proposals it generates, but these can be filtered by path length and 3D distances between reactive atoms in the output page). EzMechanism is able to find the correct mechanism for 51 enzymes out of the 55 using the default number of explored configurations (1,000), and the remaining four mechanisms (which are challenging due to their number of catalytic steps: 8, 9, 10 and 11), can still be found by increasing the number of explored configurations to 10,000.

The aim of the third test was to evaluate the coverage of the catalytic rules by ignoring, in each calculation, rules derived uniquely from the query enzyme. This is equivalent to test whether we could predict

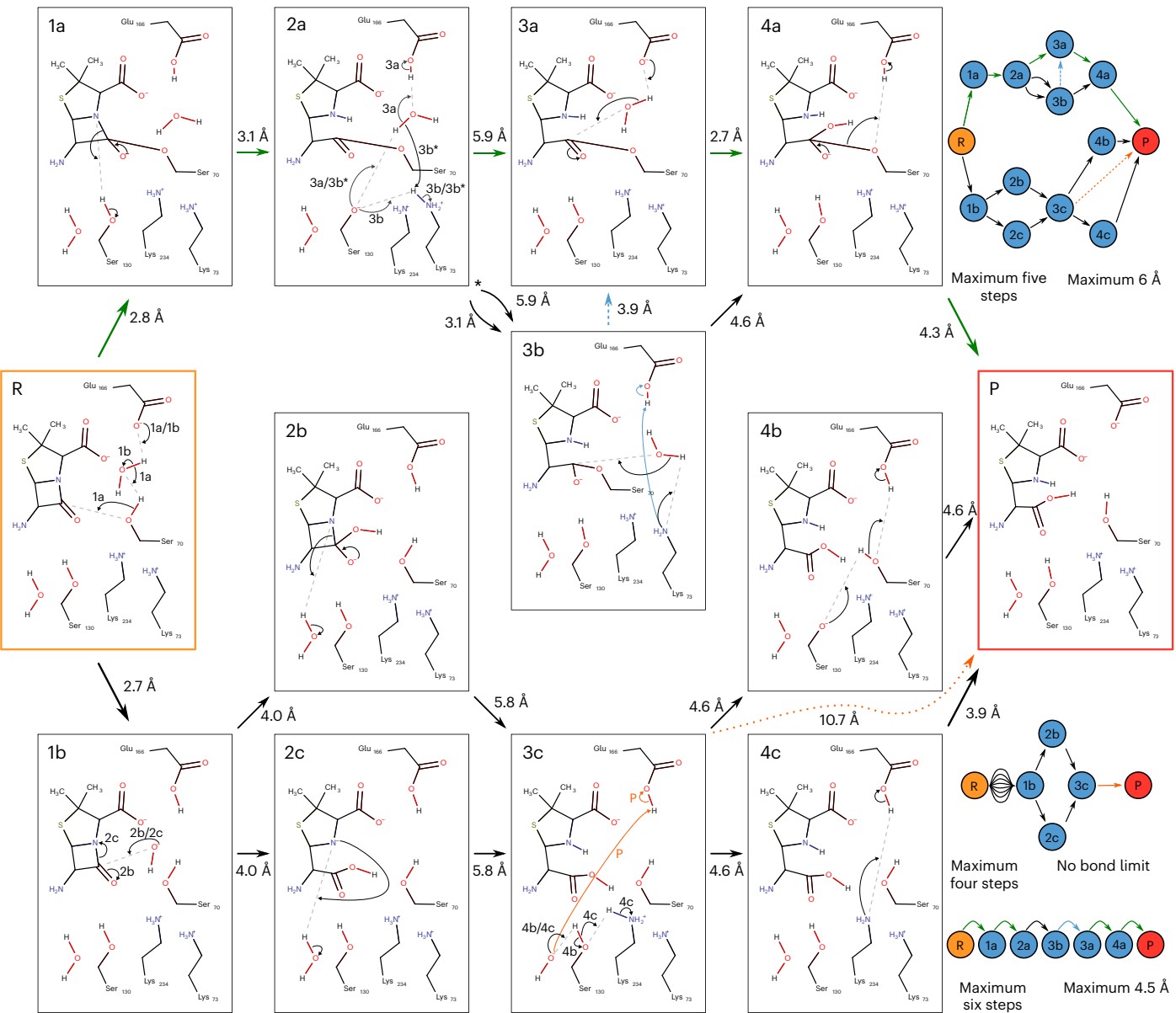

**Fig. 4 | Mechanistic paths found by EzMechanism for β-lactamase A.**
Green arrows indicate the mechanism described in the database. The graph of configurations drawn with the 2D curly arrow schemes corresponds to the graph shown on the top right. The two graphs in the bottom right represent other possible paths for the same prediction output that can be generated by considering different cut-offs for the number of steps in the path and the maximum distance of the atoms involved in the formation of new bonds.

the mechanism of the enzyme if it was not already in the database. Two versions of this test are possible: a more relaxed one that checks whether any mechanisms can be found, and a stricter version that checks whether the same exact mechanism (using the same rules, in this case also derived from other enzymes) can be found. EzMechanism can find a complete mechanism for 28 of the 55 enzymes under study, when own rules are hidden, but the exact same mechanism of the database in only 13 of these cases. The relevance of these findings is considered in the discussion.

## Validation using enzymes not annotated in M-CSA

To complement the above validation and to show how EzMechanism can be applied to enzymes that are not currently in the database, we tested the software against six enzymes, whose mechanisms have been recently studied, using their PDB structures as the starting point. To prepare and run these calculations, we used the EzMechanism public web user interface, as available to every registered user, through the M-CSA (see documentation and Supplementary Information for details). The full results of this validation are also accessible in www.ebi.ac.uk/thornton-srv/m-csa/EzMechanism/ and summarized in the Supplementary Information.

EzMechanism can find the correct mechanism (among other suggested possibilities) for four of these six enzymes. For one of the failing enzymes (PET hydrolase), the correct mechanism can still be generated if the substrate is changed slightly (in atoms two bonds away from the reaction centers, details in the Supplementary Information). The results for these enzymes reinforce what we learned in the previous validation regarding the strengths and weaknesses of EzMechanism. In particular, we found the coverage of the rules to be the main limiting factor in the failing cases. On the positive side, EzMechanism always suggests other mechanistic possibilities not considered in the referenced papers, in some cases taking into account data coming from

unrelated enzymes (in terms of fold and/or overall reaction), which would be difficult to find by other means.

One interesting example is that of COVID protease. This enzyme uses a Cys-His dyad to hydrolyze an amide bond adjacent to a Ser residue. EzMechanism can find several mechanistic paths (differing mostly in the sequence of the protonation states) that use those two residues and where Cys is the main nucleophile. However, an additional group of mechanisms is suggested where the nucleophile is instead the side chain of the serine of the substrate. Indeed, there is an enzyme annotated in M-CSA where a similar reaction occurs, leading to the formation of an oxyoxazolidine cyclic intermediate (M:225, adenosylmethionine decarboxylase, which is synthesized as a proenzyme and becomes active after a self-maturation nonhydrolytic amide cleavage). Covid protease (EC:3.4.22.69, CATH:2.40.10.10/1.10.1840.10) and adenosylmethionine decarboxylase (EC:4.1.1.50, CATH:3.60.90.10) differ both in their overall catalyzed reaction and in their structural fold.

### Test case of β-lactamase A

β-Lactamases are bacterial enzymes that hydrolyze the β-lactam ring present in penicillin and similar antibiotics, conferring resistance to these molecules. Here, we use a class A β-lactamase to exemplify how EzMechanism can be used to explore possible mechanisms.

We prepared a run of EzMechanism based on the PDB 1TEM structure[19] of the *E. coli* β-lactamase, where a substrate analog is covalently linked to Ser70, suggesting a nucleophilic role for this residue in the mechanism. Indeed, the accepted reaction mechanism, as described in the M-CSA entry for this enzyme (M-CSA ID:2) involves the creation of this covalent intermediate with the help of Glu166 and a bridging water[20]. This is followed by the cleavage of the four-membered lactam ring and a nucleophilic attack by a water molecule, which ultimately leads to the collapse of the covalent intermediate. This mechanism is identified by the green arrows in Fig. 4.

The calculation on this active site was limited to 1,000 explored configurations (configurations checked against the rules) and generated a total of 9,855 configurations. This result is typical for other enzymes we tested, which highlights the complexity and vastness of the chemical space available to enzymes. Although all these configurations are theoretically possible, most of them will not be visited by the enzyme, because they are not energetically favorable. There are computational methods able to calculate the relative energy of the enzyme states along a reaction path, but they are still too computationally expensive to be used at such a large scale. In lieu of the energy, and to facilitate the interpretation of these results, the output page of EzMechanism can use the length of the newly formed bonds (considering the positions of atoms in the PDB structure) and the number of steps in the path to filter out less productive regions of the chemical space.

Figure 4 shows the configurations and paths linking reactants to products that involve a maximum of five steps and distances shorter than 6 Å (graph in upper right corner. For simplicity, we are hiding conformation steps that involve the carboxylic group of the substrate). The mechanism described for this enzyme in the database (highlighted in green) is one of the possibilities within these criteria. There is an almost identical path, which only differs in two steps and one configuration (2a → 3b → 4a, instead of 2a → 3a → 4a). In the alternative case, Ser130 is reprotonated by Lys73 (either directly or through a water molecule), and then Lys73 is the nucleophilic base in the following step, rather than Glu166. Although the first of these steps involves closer atoms when using Lys73 (2a → 3b; 3.1 Å) rather than Glu166 (2a → 3a; 5.9 Å), this is not the case for the second step (3b → 4a; 4.6 Å versus 3a → 4a; 2.7 Å). After investigating the graph of configurations further, with different filters, we found that an additional step transforming 3b into 3a (proton transfer between Lys73 and Glu166) might present a better mechanism alternative, in terms of the distances between new bonds. This alternative mechanism has six steps and is shown in the bottom left corner of Fig. 4.

Finally, there are a couple of alternative paths that start with the deprotonation of a water molecule (R → 1b), including one possible path with only four steps involving a step with two very distant species, represented by the orange arrow. This four-step path cannot be completely excluded based on the distance criterion, because it involves one hydroxide molecule, which might move in the active site, or be substituted by other water molecules. As for other paths starting with configuration 1b and involving hydroxide, their feasibility can only be ascertained for sure with other methods, such as energy calculations. In other β-lactamase classes with mechanisms involving hydroxide molecules, these are typically stabilized by metal ions.

## Discussion

EzMechanism is a knowledge-based approach to study enzyme reaction mechanisms. It is a method able to generate mechanistic hypotheses for a given 3D active site in an automated way. EzMechanism has some important advantages when compared with other ways of producing mechanism proposals (which consist mostly of literature searches and human expertise). First, the program rule set is derived from the mechanisms of hundreds of enzymes belonging to different EC classes and structural superfamilies, which ensures good coverage of the chemical space and surpasses what most humans can recall. Second, because rule matching is purely based on local chemistry at the step level, the program does not limit the search to similar (evolutionarily related) enzymes or enzymes that catalyze the same overall reaction. Third, EzMechanism makes sure that combinations of rules are searched almost exhaustively (guided by the prioritization algorithm), which might alert the user to paths not previously considered. Finally, all rules and generated catalytic steps link back to the M-CSA entries and the original literature that were used to create them. This facilitates the comparison of the mechanism under study with the available literature, and the integration of the new studies with existing knowledge.

EzMechanism is, however, not yet a complete solution to predicting enzyme mechanisms ab initio. While the search and prioritization algorithms are able to find the correct mechanisms even when the chemical space is large, they are limited by the coverage of the catalytic rules. When excluding from the prediction rules exclusively seen in the enzyme under study, the software can only find a mechanism in about 50% of the cases, and the correct mechanism (defined as exactly the one annotated in the database) in about 25%. These should be the expected success rates of EzMechanism to find a complete mechanism, when applied to enzymes unrelated to the enzymes in M-CSA. Currently, M-CSA covers 65% of all EC subclasses (the third level of the EC classification) and 84% of the subsubclasses with a PDB structure (enzymes that share the same subsubclass typically follow the same mechanism and have a similar substrate). Even for enzymes for which EzMechanism cannot find a complete mechanism, it may still find some of the catalytic steps around the reactants or products configurations, together with the information about the original literature and enzymes that catalyze similar steps.

In future versions of the software, we aim to improve the rule set coverage by adding more mechanisms to the database, by including radical reactions in the dataset and by tweaking how the chemical information is codified in the reaction SMARTS patterns to make the rules more generic. Two additional ways to increase the number of rules might involve the manual identification and incorporation of basic rules of organic chemistry, as well as the addition of datasets commonly used for the prediction of reactions in organic syntheses[21].

By facilitating the study of enzyme mechanisms, EzMechanism might be useful to other studies and applications where this knowledge is important. For example, it can be used to evaluate the effect of catalytic residue mutations on the mechanism, which should help in the understanding of enzyme evolution, enzyme associated diseases and the design of new enzyme functions. Additionally, the software

can be used to identify potential enzyme reactions when a substrate, native or otherwise, binds in an active site.

In an ideal world, computers would be able to predict how enzymes work and the reactions they perform ab initio, based solely on their sequences. This is a hefty goal, but one that becomes more urgent as the number of available sequences and structures skyrockets. Just recently, the number of experimentally derived structures available in the PDB[22] surpassed the 200,000 and Alphafold[23] has released over 200 million predicted structures[24]. As other computational methods in the fields of bioinformatics and computational chemistry keep developing (Alphafold, Rosetta[25], QM/MM[11,12], molecular dynamics[26] and molecular docking[27], among others), we see the current and future versions of EzMechanism as a crucial cog in this ideal vision of the future, as a way to codify and apply existing chemical knowledge to make predictions.

## Online content

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

## Methods

### The M-CSA database

The M-CSA is a manually curated database of catalytic sites and enzyme mechanisms that is freely available at www.ebi.ac.uk/thornton-srv/m-csa/. Currently, M-CSA contains detailed annotations on the enzyme reaction mechanisms of 734 enzymes, including the curly arrow diagrams of 3,238 catalytic steps. For the purposes of this paper, from which we have excluded radical reactions, we have used the annotations of 691 enzymes and 2,925 catalytic steps. The data in M-CSA is unique in its scope and breadth, and it has been used over the years by ourselves and others to understand enzyme function and evolution[28–31]. Notably, we highlight the work of J. Anderson and colleagues[14], who used M-CSA data to test a new program they created that is able to perform multi-step chemical reactions in silico using graph transformation. For this, and independently from the work described in the present paper, they built a set of catalytic rules that are conceptually similar to our own 'single-step' rules. Last, they used their software and created rules to find hypothetical mechanisms (that is without being mapped to a particular 3D active site) for reactions in RHEA[32], a database of chemical reactions.

### Programming details

The code written for this project is integrated with the M-CSA codebase and uses most of the same technologies as the website, which is implemented in the python Django Web Framework (www.djangoproject.com) and uses a PostgreSQL database (www.postgresql.org). The python programming language was used for the code that extracts the chemical rules from the curly arrow diagrams and the search algorithm that tests these rules against the active site configurations. This code makes extensive use of the RDKit Cheminformatics package (www.rdkit.org) to, among other things, manipulate molecule objects, convert molecules into SMARTS and identifying matches between the chemical rules and active site configurations. We used the python library NetworkX[33] to create and manipulate the graphs used during the creation of rules for merging all the curly arrow data, and during the mechanistic search to store all the generated active site configurations and reaction paths.

The webpages used to submit new searches and analyze the results use the Django template language to generate the final html pages. The ChemAxon MarvinJS plugin v.17.15.0 (www.chemaxon.com) is used in the input webpage to draw the 2D scheme of the active site in the reactants and products configurations. The results page uses Cytoscape JS[34] to show the graph with the computed mechanistic reaction paths. Custom Javascript code was written to filter and control the data presented in this graph.

### Creation of the rules of enzyme catalysis

The catalytic rules created in this work are based on the curly arrow diagrams of the mechanistic steps in M-CSA (Fig. 1a). These data, which include all the pictured atoms and bonds, as well as the curly arrows that indicate the formation and cleavage of bonds, are stored internally as ChemAxon Marvin files (.mrv extension). Marvin files follow a custom XML schema, which we parse using a custom python script, using the 'lxml' python package. Molecules in these files are first recreated as rdkit Mol objects and then converted to a SMARTS string. The SMARTS strings for all molecules in the diagram are then saved in the M-CSA database, together with the information about the curly arrows for each step.

In most cases, each catalytic step is associated with a catalytic rule. This is the case for the examples shown in Fig. 1, and it is also what happens when we apply these rules in the search algorithm, that is, each match of a rule will generate a new configuration that corresponds to a new proposed mechanistic step. This might suggest a straightforward way to create catalytic rules, in which each step in the database will be used to create a single catalytic rule. We did not follow this route for a couple of reasons: (1) there might be independent chemical activities

in the same step, such as two proton transfers (not coupled), which might be better described by two catalytic rules and (2) rules created in this way might be overly rigid, in particular when there is a long chain of curly arrows. For example, from step A of Fig. 1, we know that a water molecule can be activated to perform a nucleophilic attack on phosphate by a histidine. From step B, on the other hand, we learn that a water performing a similar nucleophilic attack can also be activated by a lysine. By using information such as this and combining information from different steps we were able to generate more generic rules.

To mix the information of different steps in the database, we start by representing each chain of curly arrows as a directed graph. Each curly arrow is represented by a node and edges indicate that the curly arrows act sequentially on the same atom or bond. The first and last curly arrows in each chain of arrows are labeled as such. Each node (curly arrow) is defined by the atoms or bonds that it touches and surrounding atoms up to two bonds away. After repeating this process for each step, all the generated graphs are merged by combining nodes (curly arrows) that carry the same information. Figure 2a shows a combined curly arrow graph for the pictured catalytic steps.

Starting with the combined graph that contains information of all the curly arrows in the database, rules are created by following this process: (1) find all the simple paths between 'starting' nodes and 'ending' nodes, as well as circular paths, that have ten or fewer nodes. (2) Keep all the paths with curly arrows coming from the same step but remove mixed paths (paths with steps coming from different enzymes) with more than six curly arrows. This choice of cut-off guarantees that the number of generated mixed-step rules is manageable, and roughly similar to the number of single-step rules. (3) For each path, build all the catalytic rules that might be generated by the merging of its curly arrows. (4) Reverse the reactants and products of the generated rules to also obtain the respective reverse rules.

The rules created in this way are saved to the database as reaction SMARTS and each rule is linked back to the enzymes and mechanistic steps that were used to create it.

### Input preparation and webserver pages

We developed a web user interface that any registered user of M-CSA can use to submit EzMechanism calculations. The submission is a multi-step process that captures all the information required by the software: (1) choice of a PDB structure; (2) selection of the catalytic residues; (3) definition of the substrates and cofactors and their mapping to the PDB structure and (4) definition of the overall reaction in the context of the active site, using a 2D schematic representation. The web input page contains a 3D viewer of the protein, that can be focused on the active site to help the selection of the correct catalytic residues and ligands. To facilitate the mapping of the native substrate to the correct PDB ligand, the page shows the distance of every ligand in the PDB to the active site and the similarity of each ligand to the native substrate. After a ligand is selected, a visual aid shows the superposition of both molecules' maximum-common substructure. Finally, the drawing of the reactants and products configurations is done using the MarvinJs plugin, while the page verifies that both configurations are balanced, and that the atom-to-atom mapping is correct.

A more detailed explanation of how to submit EzMechanism calculations is accessible at https://www.ebi.ac.uk/thornton-srv/m-csa/EzMechanism/documentation and in the Supplementary Information.

### The search algorithm setup

EzMechanism uses diverse kinds of information and molecular representations to navigate through the catalytic chemical space:

(1) Balanced 2D schemes of the active site that are first created for the reactant and product configurations and later for all the configurations generated during the mechanistic search

(2) A 3D representation of the active site, which is used to check distances of formed bonds when creating new active site configurations, as taken from the PDB file selected in the input

(3) A SMILES string representation of each active site configuration used to define a unique identifier

Internally, the search algorithm uses a list of RDKit Mol objects (one for each molecule in the active site, including protein residues, the substrate and cofactors) to store the information associated with each configuration. Each Mol object contains two sets of coordinates that correspond to the 2D and 3D representations of the active site, and the molecular topology information necessary to create the SMILES identifier. All heavy atoms in the active site are labeled with a unique identifier (atom map) so that the algorithm can distinguish between identical molecules (such as two identical catalytic residues, for example). Hydrogens atoms, on the contrary, are deemed equivalent, so although the protonation state of molecules is considered, the origin of their protons is not.

The 3D coordinates attributed to the reactants' configuration are taken from the selected PDB file. Molecules in the internal representation of the active site (including catalytic residues, cofactors and substrates) are first mapped to the correct PDB residues, according to the mapping defined by the user. The software superimposes the common parts of the cognate ligand onto the PDB ligand and then optimizes the 3D coordinates of the remaining atoms of the cognate ligand using the MMFF94 force field, as implemented in RDKit.

Finally, the 2D schemes of the active site file representing the reactants and products configurations of the reaction are used to identify what are the reaction centers of the overall enzyme reaction, that is, atoms that are involved in the cleavage and formation of bonds. This information will be used in the prioritization algorithm, which will favor chemical steps that involve the overall reaction centers.

### Search and optimizations in the search algorithm

The starting point for the search of possible mechanisms is the active site configuration of the reactants. Using the RDkit rdChemReactions module, the software checks every catalytic rule against this configuration and then generates all possible new configurations based on the matches. These new configurations are possible 'first intermediates' of the reaction path, and the rules that matched correspond to potential 'first steps' of the mechanism. This process of rule matching is repeated for the products configuration, which, in a reverse manner, leads to the generation of potential 'last steps' of the mechanism. The process then continues in an iterative way for the newly generated configurations, leading to an increasing number of configurations and potential mechanistic steps, which represent the chemical space of reactions available for this active site. Ideally, at the end of the run, there will be one or more mechanistic paths that connect the reactants configuration with the products configuration.

While exploring the chemical space around the reactants and products, many possible configurations are generated and some of these are similar or identical to each other. These similar configurations arise from three types of situation: (1) a given catalytic rule might match two identical catalytic residues or molecules in the active site, such as a rule that protonates a carboxylic group using water, which will match two Asp residues. In this case, two new similar configurations are possible, where each one of the Asps is protonated in turn. (2) A single match to the same exact molecule might lead to several new configurations when the matching molecule contains equivalent atoms. A trivial example is a proton transfer from a protonated Lys side chain that generates three new configurations, one for each proton bound to the terminal nitrogen. (3) The same or similar configurations might be found by starting from different configurations and following different rules and mechanistic paths, just by chance. In all these situations, the software needs to decide

what should be considered a new or already seen configuration. For this purpose, the algorithm considers that all hydrogen atoms are indistinguishable while heavy atoms are unique (by giving them a unique atom map identifier). This rule avoids the proliferation of superfluous active site configurations created by permutations of hydrogen atoms, as in example (2) above, while considering that residues or other molecules are always unique, even if there is more than one type of the molecule in the active site, as in example (1).

In a typical run, the number of potential active site configurations that can be generated (which represents the size of the available chemical space) is too big to be explored in an exhaustive way. For this reason, we have developed a prioritization algorithm that guides the search toward configurations that are more relevant for the mechanistic path. The prioritization algorithm uses a score that favors: (1) steps where new bonds are formed between close atoms (using 3D coordinates) rather than distant atoms; (2) steps where new or cleaved bonds involve the reaction centers of the overall enzyme reaction and (3) by virtue of Dijkstra's algorithm[35], which tries to find the shortest path between nodes and computes the overall distance as the sum of scores of every edge (step), steps that are closer to one of the starting configurations are also favored (details in the Supplementary Information).

As described in the first paragraph of this section, the software starts the search from the reactant's configuration followed by the product's configuration. In the subsequent iterations, the search alternates between the two sides of the mechanism, so, rather than a reactants-to-products direction, the search is done from both sides simultaneously (taking advantage of the reverse rules) with the aim of find overlapping configurations that join the two sides. This bidirectional search is used to make the search notably more efficient. In a one-directional search, the number of configurations ($c$) to explore grows exponentially with the number of mechanistic steps ($s$), $c = \sum_{i=1}^{s} n^i$, where $n$ is the number of rule matches per configuration (assumed here to be always the same for simplification). With the bidirectional search, however, each side must only explore half of the steps and the formula for the total number of explored configurations becomes $c = 2 \times \sum_{i=1}^{s/2} n^i$. For a six-step mechanism with $n = 8$, for example, the number of generated configurations goes down from roughly 300,000 to around 1,000, when comparing the two approaches.

In initial versions of the software, we noticed that most of the computational resources (both computer processing unit and memory) were being spent on creating and saving new configurations and checking the rules against the configurations. We have implemented the following changes to limit the cost of these operations: (1) instead of representing each active site configuration as a single object they are represented as a list of normalized molecules, meaning that the representation of equivalent molecules in computer memory is shared among configurations; (2) the normalization of molecules across configurations allows for the caching of the rule-matching calculations and (3) the rule-matching function of RDKit requires the creation of a list of molecules with the same length as the number of molecules in the rule. Instead of just creating all permutations of a given length with all the molecules in the active site, the software first runs a substructure match between each molecule and part of the rule and the permutations are just created for the matching molecules. The results of the substructure matching are also cached for efficiency.

### Webpage for the analysis of the output

The output of EzMechanism calculations may contain an overwhelming number of active site configurations and reaction steps. To facilitate the interpretation of these results, we developed a custom webpage (the output of the validation calculations is shown using this page and is available at www.ebi.ac.uk/thornton-srv/m-csa/EzMechanism/). The output page contains three main panels, one showing the graph of configurations and steps, another with buttons used to filter the graph and show information about the catalytic rule of the selected reaction

step and a third showing the 2D diagrams of selected configurations. If the mechanism prediction is done for a database entry that already contains a mechanism, the mechanism already in the database is shown in a fourth panel for comparison purposes. A detailed description of the output page and its capabilities is available in the Supplementary Information, and in the documentation pages of EzMechanism (https://www. ebi.ac.uk/thornton-srv/m-csa/EzMechanism/documentation_output).

## Reporting summary

Further information on research design is available in the Nature Portfolio Reporting Summary linked to this article.

## Data availability

Data used during this study, in particular the machine-readable files of the catalytic steps, are available for download at the M-CSA website (https://www.ebi.ac.uk/thornton-srv/m-csa/download/). The generated 'rules of catalysis' and the validation results can be browsed using the M-CSA web interface at https://www.ebi.ac.uk/thornton-srv/m-csa/rules/, and https://www.ebi.ac.uk/thornton-srv/m-csa/EzMechanism/, respectively. These data are also available to download as flat files in https://www.ebi.ac.uk/thornton-srv/m-csa/media/ezmechanism_data_share/2023_nat_met_rules.tar.xz and https://www.ebi.ac.uk/thornton-srv/m-csa/media/ezmechanism_data_share/2023_nat_met_validation.tar, and are archived in https://zenodo.org/record/7957138.

## Code availability

EzMechanism code is integrated into the M-CSA codebase. Code specific to EzMechanism is available at https://zenodo.org/record/7993958, and the complete source code for the M-CSA website is available upon request.

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

## Acknowledgements

This work would not be possible without the contribution of all past M-CSA curators and maintainers. In particular, we highlight the contribution of G. Holiday and N. Furnham, as well as of the last group of curators: M. Babić, N. Marson, Y. Abeje and T. Nguyen. A complete list of people involved in the creation and curation of the M-CSA database is available at https://www.ebi.ac.uk/thornton-srv/m-csa/about/. We also thank R. Laskowski and G. Hann for helpful discussions and suggestions. This work was funded by the European Molecular Biology Laboratory (A.J.M.R., J.M.T., N.B. and J.M.T.) and the European Molecular Biology Laboratory International PhD Programme (I.G.R.). The funders had no role in study design, data collection and analysis, decision to publish or preparation of the manuscript.

## Author contributions

All authors were involved in the conception and design of the work, as well as the revision of the manuscript. A.J.M.R. developed the code, performed the validation of the software and wrote the first draft of the manuscript.

## Funding

## Competing interests

The authors declare no competing interests.

## Additional information

**Correspondence and requests for materials** should be addressed to Antonio J. M. Ribeiro or Janet M. Thornton.

# Reporting Summary

Please do not complete any field with "not applicable" or n/a. Refer to the help text for what text to use if an item is not relevant to your study.
For final submission: please carefully check your responses for accuracy; you will not be able to make changes later.

## Statistics

For all statistical analyses, confirm that the following items are present in the figure legend, table legend, main text, or Methods section.

| n/a | Confirmed | |
|---|---|---|
| ☒ | ☐ | The exact sample size ($n$) for each experimental group/condition, given as a discrete number and unit of measurement |
| ☒ | ☐ | A statement on whether measurements were taken from distinct samples or whether the same sample was measured repeatedly |
| ☒ | ☐ | The statistical test(s) used AND whether they are one- or two-sided<br>*Only common tests should be described solely by name; describe more complex techniques in the Methods section.* |
| ☒ | ☐ | A description of all covariates tested |
| ☒ | ☐ | A description of any assumptions or corrections, such as tests of normality and adjustment for multiple comparisons |
| ☒ | ☐ | A full description of the statistical parameters including central tendency (e.g. means) or other basic estimates (e.g. regression coefficient) AND variation (e.g. standard deviation) or associated estimates of uncertainty (e.g. confidence intervals) |
| ☒ | ☐ | For null hypothesis testing, the test statistic (e.g. $F$, $t$, $r$) with confidence intervals, effect sizes, degrees of freedom and $P$ value noted<br>*Give P values as exact values whenever suitable.* |
| ☒ | ☐ | For Bayesian analysis, information on the choice of priors and Markov chain Monte Carlo settings |
| ☒ | ☐ | For hierarchical and complex designs, identification of the appropriate level for tests and full reporting of outcomes |
| ☒ | ☐ | Estimates of effect sizes (e.g. Cohen's $d$, Pearson's $r$), indicating how they were calculated |

*Our web collection on statistics for biologists contains articles on many of the points above.*

## Software and code

Policy information about availability of computer code

| Data collection | N/A |
|---|---|
| Data analysis | The EzMechanism code is available at https://zenodo.org/record/7993958 and the M-CSA website codebase can be made available upon request. |

For manuscripts utilizing custom algorithms or software that are central to the research but not yet described in published literature, software must be made available to editors and reviewers. We strongly encourage code deposition in a community repository (e.g. GitHub). See the Nature Portfolio guidelines for submitting code & software for further information.

## Data

Policy information about availability of data

All manuscripts must include a data availability statement. This statement should provide the following information, where applicable:
- Accession codes, unique identifiers, or web links for publicly available datasets
- A description of any restrictions on data availability
- For clinical datasets or third party data, please ensure that the statement adheres to our policy

Data used during this study, in particular the machine-readable files of the catalytic steps, are available for download at the M-CSA website (https://www.ebi.ac.uk/thornton-srv/m-csa/download/).
The generated "rules of catalysis" and the validation results can be browsed using the M-CSA web interface at https://www.ebi.ac.uk/thornton-srv/m-csa/rules/, and https://www.ebi.ac.uk/thornton-srv/m-csa/EzMechanism/, respectively. These data are also available to download as flat files in https://www.ebi.ac.uk/thornton-srv/m-csa/media/ezmechanism_data_share/2023_nat_met_rules.tar.xz and https://www.ebi.ac.uk/thornton-srv/m-csa/media/ezmechanism_data_share/2023_nat_met_validation.tar, and are archived in https://zenodo.org/record/7957138.

## Research involving human participants, their data, or biological material

Policy information about studies with human participants or human data. See also policy information about sex, gender (identity/presentation), and sexual orientation and race, ethnicity and racism.

| | |
|---|---|
| Reporting on sex and gender | N/A |
| Reporting on race, ethnicity, or other socially relevant groupings | N/A |
| Population characteristics | N/A |
| Recruitment | N/A |
| Ethics oversight | N/A |

Note that full information on the approval of the study protocol must also be provided in the manuscript.

# Field-specific reporting

Please select the one below that is the best fit for your research. If you are not sure, read the appropriate sections before making your selection.

☒ Life sciences    ☐ Behavioural & social sciences    ☐ Ecological, evolutionary & environmental sciences

For a reference copy of the document with all sections, see nature.com/documents/nr-reporting-summary-flat.pdf

# Life sciences study design

All studies must disclose on these points even when the disclosure is negative.

| | |
|---|---|
| Sample size | N/A |
| Data exclusions | N/A |
| Replication | N/A |
| Randomization | N/A |
| Blinding | N/A |

# Behavioural & social sciences study design

All studies must disclose on these points even when the disclosure is negative.

| | |
|---|---|
| Study description | |
| Research sample | |
| Sampling strategy | |
| Data collection | |
| Timing | |
| Data exclusions | |
| Non-participation | |
| Randomization | |

# Ecological, evolutionary & environmental sciences study design

All studies must disclose on these points even when the disclosure is negative.

| | |
|---|---|
| Study description | |
| Research sample | |
| Sampling strategy | |
| Data collection | |
| Timing and spatial scale | |
| Data exclusions | |
| Reproducibility | |
| Randomization | |
| Blinding | |

Did the study involve field work?  ☐ Yes  ☐ No

## Field work, collection and transport

| | |
|---|---|
| Field conditions | |
| Location | |
| Access & import/export | |
| Disturbance | |

# Reporting for specific materials, systems and methods

We require information from authors about some types of materials, experimental systems and methods used in many studies. Here, indicate whether each material, system or method listed is relevant to your study. If you are not sure if a list item applies to your research, read the appropriate section before selecting a response.

### Materials & experimental systems

| n/a | Involved in the study |
|-----|------------------------|
| ☒ | Antibodies |
| ☒ | Eukaryotic cell lines |
| ☒ | Palaeontology and archaeology |
| ☒ | Animals and other organisms |
| ☒ | Clinical data |
| ☒ | Dual use research of concern |
| ☒ | Plants |

### Methods

| n/a | Involved in the study |
|-----|------------------------|
| ☒ | ChIP-seq |
| ☒ | Flow cytometry |
| ☒ | MRI-based neuroimaging |

## Antibodies

| | |
|---|---|
| Antibodies used | |
| Validation | |

# Eukaryotic cell lines

Policy information about cell lines and Sex and Gender in Research

| | |
|---|---|
| Cell line source(s) | |
| Authentication | |
| Mycoplasma contamination | |
| Commonly misidentified lines<br>(See ICLAC register) | |

# Palaeontology and Archaeology

| | |
|---|---|
| Specimen provenance | |
| Specimen deposition | |
| Dating methods | |

☐ Tick this box to confirm that the raw and calibrated dates are available in the paper or in Supplementary Information.

| | |
|---|---|
| Ethics oversight | |

Note that full information on the approval of the study protocol must also be provided in the manuscript.

# Animals and other research organisms

Policy information about studies involving animals; ARRIVE guidelines recommended for reporting animal research, and Sex and Gender in Research

| | |
|---|---|
| Laboratory animals | |
| Wild animals | |
| Reporting on sex | |
| Field-collected samples | |
| Ethics oversight | |

Note that full information on the approval of the study protocol must also be provided in the manuscript.

# Clinical data

Policy information about clinical studies

All manuscripts should comply with the ICMJE guidelines for publication of clinical research and a completed CONSORT checklist must be included with all submissions.

| | |
|---|---|
| Clinical trial registration | |
| Study protocol | |
| Data collection | |
| Outcomes | |

# Dual use research of concern

Policy information about dual use research of concern

## Hazards

Could the accidental, deliberate or reckless misuse of agents or technologies generated in the work, or the application of information presented in the manuscript, pose a threat to:

| No | Yes | |
|----|-----|---|
| ☒ | ☐ | Public health |
| ☒ | ☐ | National security |
| ☒ | ☐ | Crops and/or livestock |
| ☒ | ☐ | Ecosystems |
| ☒ | ☐ | Any other significant area |

## Experiments of concern

Does the work involve any of these experiments of concern:

| No | Yes | |
|----|-----|---|
| ☒ | ☐ | Demonstrate how to render a vaccine ineffective |
| ☒ | ☐ | Confer resistance to therapeutically useful antibiotics or antiviral agents |
| ☒ | ☐ | Enhance the virulence of a pathogen or render a nonpathogen virulent |
| ☒ | ☐ | Increase transmissibility of a pathogen |
| ☒ | ☐ | Alter the host range of a pathogen |
| ☒ | ☐ | Enable evasion of diagnostic/detection modalities |
| ☒ | ☐ | Enable the weaponization of a biological agent or toxin |
| ☒ | ☐ | Any other potentially harmful combination of experiments and agents |

# Plants

| Seed stocks | |
|---|---|
| Novel plant genotypes | |
| Authentication | |

# ChIP-seq

## Data deposition

☐ Confirm that both raw and final processed data have been deposited in a public database such as GEO.

☐ Confirm that you have deposited or provided access to graph files (e.g. BED files) for the called peaks.

| Data access links
May remain private before publication. | |
|---|---|
| Files in database submission | |
| Genome browser session
(e.g. UCSC) | |

## Methodology

| Replicates | |
|---|---|
| Sequencing depth | |
| Antibodies | |
| Peak calling parameters | |
| Data quality | |
| Software | |

# Flow Cytometry

## Plots

Confirm that:

- ☐ The axis labels state the marker and fluorochrome used (e.g. CD4-FITC).
- ☐ The axis scales are clearly visible. Include numbers along axes only for bottom left plot of group (a 'group' is an analysis of identical markers).
- ☐ All plots are contour plots with outliers or pseudocolor plots.
- ☐ A numerical value for number of cells or percentage (with statistics) is provided.

## Methodology

| | |
|---|---|
| Sample preparation | |
| Instrument | |
| Software | |
| Cell population abundance | |
| Gating strategy | |

☐ Tick this box to confirm that a figure exemplifying the gating strategy is provided in the Supplementary Information.

# Magnetic resonance imaging

## Experimental design

| | |
|---|---|
| Design type | |
| Design specifications | |
| Behavioral performance measures | |

| | |
|---|---|
| Imaging type(s) | |
| Field strength | |
| Sequence & imaging parameters | |
| Area of acquisition | |

Diffusion MRI   ☐ Used   ☐ Not used

## Preprocessing

| | |
|---|---|
| Preprocessing software | |
| Normalization | |
| Normalization template | |
| Noise and artifact removal | |
| Volume censoring | |

## Statistical modeling & inference

| | |
|---|---|
| Model type and settings | |
| Effect(s) tested | |

Specify type of analysis:   ☐ Whole brain   ☐ ROI-based   ☐ Both

Statistic type for inference

(See Eklund et al. 2016)

Correction

## Models & analysis

n/a | Involved in the study
☐ ☐ Functional and/or effective connectivity
☐ ☐ Graph analysis
☐ ☐ Multivariate modeling or predictive analysis

Functional and/or effective connectivity

Graph analysis

Multivariate modeling and predictive analysis

