## [Peer Review File · Nature Methods]

Peer Review Information

Manuscript Title: EzMechanism: An Automated Tool to Propose Catalytic Mechanisms of Enzyme Reactions

Corresponding author name(s): António Ribeiro

Editorial Notes: n/a

Reviewer Comments & Decisions:

Decision Letter, initial version:

Dear António,

Your Article entitled "EzMechanism: An Automated Tool to Propose Catalytic Mechanisms of Enzyme Reactions" has now been seen by 3 reviewers, whose comments are attached. While they find your work of some potential interest, they have raised concerns which in our view are sufficiently important that they preclude publication of the work in Nature Methods.

We will consider looking at a revised manuscript only if further experimental data allow you to address all the major criticisms of the reviewers (unless, of course, something similar has by then been accepted at Nature Methods or appeared elsewhere). This includes submission or publication of a portion of this work somewhere else.

The required new experiments and data include, but are not limited to the inclusion of a blind test case to show the approach actually works on unknown/new enzymes. Additionally, it would be important to provide EzMechanism as a broadly usable tool with detailed usage instructions and examples/tutorials. We hope you understand that until we have read the revised paper in its entirety we cannot promise that it will be sent back for peer-review.

If you are interested in revising this manuscript for submission to Nature Methods in the future, please contact me to discuss your appeal before making any revisions. Otherwise, we hope that you find the reviewers' comments helpful when preparing your paper for submission elsewhere.

Sincerely,

Arunima

Arunima Singh, Ph.D.
Senior Editor
Nature Methods

Reviewers' Comments:

Reviewer #1:

Remarks to the Author:

The authors describe a computer algorithm to predict possible reaction mechanisms based on inspection of the identity and arrangement of possible catalytic residues in an enzyme structure. Analysis is based on the development of single reaction step rules that define the movement of electrons during an enzyme-catalyzed reaction and then the application of these rules in a search for possible mechanisms. Although this is a daunting task and this manuscript represents just the beginning of ongoing development, the authors are encouraged by the relatively small number of biocatalytic rules. Moreover, they show that the algorithm succeeds most of the time when given enzyme structures with known reaction mechanism. There are numerous limitations such as the need to consider conformational variability and energy landscape, but the authors are well aware of the problems that lay ahead for further development. It is likely that the algorithms will also be useful to explore evolution of enzyme activity and the engineering of new activities. The manuscript is well written and honestly depicts the current state of the art and directions for improvement.

Reviewer #2:

Remarks to the Author:

The authors described a computational method to propose reaction mechanisms given the enzyme structure, substrate, product and catalytic residues. The authors used the existing M-CSA database to derive a set of catalytic rules and used these rules to enumerate possible reaction steps. This approach provides a knowledge-based way to predict reaction mechanisms and is less computationally expensive than ab initio calculations. This predictive algorithm is highly interpretable and this tool could be very useful to biochemists and enzymologists. However, the authors did not present a systematic approach in testing the accuracy of this predictive algorithm and therefore it is not entirely convincing that this tool produces robust results and would be broadly applicable. For example, the authors did not report blind testing for enzymes not present in the M-CSA database and it seems like the accuracy dropped significantly when rules unique to individual enzymes tested were excluded (see more below). In

addition, the authors described heuristics (atomic distances, number of steps) to traverse the vast space of possible mechanisms but did not explain why their choices of heuristics are appropriate and how changing heuristic criteria change predictions. Lastly, the author did not provide a link to the webpage for user submission (line 410: “The webpages used to submit new searches and analyze the results..” and line 459: “The submission process is done through the M-CSA website...”, but no link given); also importantly, the code package contains very limited documentation, providing no information about how to run the program. Therefore, we were not able to assess if the code can run smoothly and there are limitations on how user friendly this software would be. We recommend (in the future) making the codebase publicly available with appropriate README and tutorials to help users navigate the software.

Overall, we feel that the tool is not well enough established (strengths and limitations) and the supporting software not sufficiently developed for publication as a method.

Line 230: What is the criteria for a ligand to be similar to “cognate” substrate? If the ligand bound is a transition state analog, it may look different from the substrate, but it is similar to a reaction intermediate. How do the types of ligands present in the PDB structure affect the prediction?

Line 255: How unusual is this scenario? What is the frequency of observing this across all mechanistic steps in M-CSA?

Line 258: The authors can perform blind testing by removing these test enzymes from MCSA to derive the catalytic rules and compare if this affects mechanistic predictions of the test set. Alternatively, the authors can take a recently elucidated enzyme mechanism not present in the M-CSA and use that to test the algorithm.

This is similar to what is described at line 275 but authors did not report how many mechanisms were predicted correctly.

In the more relaxed test described at line 280, removing enzyme-specific rules did make the accuracy drop by ~50%, highlighting the need to address this issue.

In the test set, the majority of enzymes have “unique rules”. First, what are these unique rules? Second, how often do enzymes exhibit unique rules across the MCSA? These information are not included but would be essential in showing how widely applicable this prediction algorithm would be.

Line 271: What is the time-complexity of this prediction algorithm as the number of explored configurations increased?

Other feedback:

The prioritization is based on distances of atoms and number of steps and authors noted the challenge of having a huge number of possible steps. Although this opens up more possibilities, would it be possible to use chemical heuristics to simplify the search if users chose to do so? For example, presence of certain cofactors as they would have a set of known mechanistic steps in the database that is the most probable, due to their chemical properties.

User friendliness and scripting interface. It seems like prediction needs to be run on an online server and users input molecules by drawing 2D schemes. In future renditions it might be useful to have a python scripting API so that users can run it in high throughput, but this depends on target users and use cases.

Reviewer #3:

Remarks to the Author:

The authors describe the generation of 'chemical' rules deduced from known enzyme mechanisms and their application in an automated tool using 3D data of an active site to propose novel and/or alternative mechanisms for enzymes. The manuscript is overall well written and understandable to enzymologists not only to chemists. The tool is of broader interest to the world of enzymology and biocatalysis and may aid the understanding and development of enzymes.

However, I have a few comments:

Major points:

I do understand the important role of M-CSA for this publication, however how this is managed leaves me puzzled that it is partly in the introduction, even explained there, later in the methods. Either reduce this part and cite the reference or use this publication as a kind of update of the 2018 NAR paper and make M-CSA more visible in title and abstract.

The part starting from line 53 should be shortened to a single paragraph, it appears partly redundant to the discussion.

Is there a test case for an enzyme of a yet unknown mechanism available to prove the capacities for hypothesis generation?

I think it would be good to have this tool available independent of a M-CSA curator account prior to publication.

Minor points:

Line 24-25: catalytic macromolecules are not abundant in genomes, genes (putatively) encoding for enzymes are.

Author Rebuttal to Initial comments

Reviewers' Comments:

Reviewer #1:

Remarks to the Author:

The authors describe a computer algorithm to predict possible reaction mechanisms based on inspection of the identity and arrangement of possible catalytic residues in an enzyme structure.

Analysis is based on the development of single reaction step rules that define the movement of electrons during an enzyme-catalyzed reaction and then the application of these rules in a search for possible mechanisms. Although this is a daunting task and this manuscript represents just the beginning of ongoing development, the authors are encouraged by the relatively small number of biocatalytic rules. Moreover, they show that the algorithm succeeds most of the time when given enzyme structures with known reaction mechanism. There are numerous limitations such as the need to consider conformational variability and energy landscape, but the authors are well aware of the problems that lay ahead for further development. It is likely that the algorithms will also be useful to explore evolution of enzyme activity and the engineering of new activities. The manuscript is well written and honestly depicts the current state of the art and directions for improvement.

We thank the reviewer for the positive and encouraging words. We agree the current method has important limitations, which we tried to make clear in the manuscript. This is a new approach to the study of enzyme mechanisms, and we plan to address some of these limitations in the future.

Reviewer #2:

Remarks to the Author:

*The authors described a computational method to propose reaction mechanisms given the enzyme structure, substrate, product and catalytic residues. The authors used the existing M-CSA database to derive a set of catalytic rules and used these rules to enumerate possible reaction steps. This approach provides a knowledge-based way to predict reaction mechanisms and is less computationally expensive than *ab initio* calculations. This predictive algorithm is highly interpretable and this tool could be very useful to biochemists and enzymologists.*

We thank the reviewer for the positive comments. We would like to clarify that although it is true that EzMechanism is much faster than *ab initio* calculations we do not think it should be seen in opposition (or competition) with those kinds of methods. Our software is able to come up with possible mechanistic pathways for a given enzyme in an automated manner and taking the literature knowledge into account. This is something that has not been done before and out of scope for *ab initio* methods. On the other hand, EzMechanism only provides indirect measures (the distance between atoms that form new bonds, and the length of the mechanistic path) to assess if the discovered paths are feasible. This is one of the main purposes of *ab initio* methods, which are able to calculate the energies of these configurations (and their transition states). Hence, we see these two approaches as complementary.

However, the authors did not present a systematic approach in testing the accuracy of this predictive algorithm and therefore it is not entirely convincing that this tool produces robust results and would be broadly applicable. For example, the authors did not report blind testing for enzymes not present in the M-CSA database and it seems like the accuracy dropped significantly when rules unique to individual enzymes tested were excluded (see more below).

After considering this feedback (and similar feedback by other reviewers), we have performed tests on 6 enzymes that do not exist in the database. The results of these tests are presented in the new version of the manuscript in the "Validation using Enzymes not annotated in the M-CSA" section, and in more detail in the SI. Results for these enzymes are also available in the M-CSA website: <https://www.ebi.ac.uk/thornton-srv/m-csa/EzMechanism/>.

In addition, the authors described heuristics (atomic distances, number of steps) to traverse the vast space of possible mechanisms but did not explain why their choices of heuristics are appropriate and how changing heuristic criteria change predictions.

The prioritisation score (formula shown in the SI) is the heuristic used to guide the search. Indeed, in the first version of the manuscript we did not discuss the rationale for the scoring function. We have expanded the discussion of the prioritisation function in the SI to explain some of the choices we made for its construction.

During the validation we verified that the current prioritisation algorithm works quite well (In all validation cases we could always find the correct mechanism among the other possibilities), so a systematic exploration of the prioritisation function parameters was not called for. We may return to this in the future, if necessary, after improving other parts of the software, such as the coverage of the rules or the ability to use modelled structures.

Lastly, the author did not provide a link to the webpage for user submission (line 410: "The webpages used to submit new searches and analyze the results.." and line 459: "The submission process is done through the M-CSA website...", but no link given);

As explained above in the response to the editor, in the first version of the manuscript, we were not planning on opening the software to other users at that moment, because the submission process was complex, requiring the use of several pages. Since this was identified by the reviewers and the editor as a crucial point, we have developed a new page that anyone can use (after registering) to submit their own queries based on a PDB structure. Instructions to do this are given in the response to the editor (including a username and password created for each reviewer) and in the new documentation page (<https://wwwdev.ebi.ac.uk/thornton-srv/m-csa/EzMechanism/documentation>).

also importantly, the code package contains very limited documentation, providing no information about how to run the program. Therefore, we were not able to assess if the code can run smoothly and there are limitations on how user friendly this software would be. We recommend (in the future) making the codebase publicly available with appropriate README and tutorials to help users navigate the software.

Two new documentation pages were created and are now available through the website, one to explain the submission process: <https://wwwdev.ebi.ac.uk/thornton-srv/m-csa/EzMechanism/documentation>; and another to explain how to use the results page: https://www.ebi.ac.uk/thornton-srv/m-csa/EzMechanism/documentation_output. The information contained in these two webpages is also available in the SI.

Overall, we feel that the tool is not well enough established (strengths and limitations) and the supporting software not sufficiently developed for publication as a method.

Line 230: What is the criteria for a ligand to be similar to "cognate" substrate? If the ligand bound is a transition state analog, it may look different from the substrate, but it is similar to a reaction intermediate. How do the types of ligands present in the PDB structure affect the prediction?

The maximum-common-substructure was used to calculate the similarity between the cognate and the PDB ligand, using the "PARITY" method (<https://doi.org/10.1016/j.str.2018.02.009>), using a cut-

off of 70%. We note that this choice of PDBs is only relevant for the validation. Users can choose any structure available in the PDB. The new user interface shows in a clear way how the superposition of cognate and PDB ligands are calculated.

As for the effect in the prediction, if the cognate and PDB ligands are very similar, the generated 3D model will be of better quality and so the distances calculated for the prioritisation algorithm will be more accurate. On the other hand, very different molecules will lead to a worse 3D model (where the substrate might be positioned in an incorrect pose, for example) and the search algorithm will not find the most productive configurations as easily.

Line 255: How unusual is this scenario? What is the frequency of observing this across all mechanistic steps in M-CSA?

Across the entire database, we identified 7 steps where a similar problem occurs (among 3238).

Line 258: The authors can perform blind testing by removing these test enzymes from MCSA to derive the catalytic rules and compare if this affects mechanistic predictions of the test set.

This is similar to what is described at line 275 but authors did not report how many mechanisms were predicted correctly. In the more relaxed test described at line 280, removing enzyme-specific rules did make the accuracy drop by ~50%, highlighting the need to address this issue.

We have not designed our validation in this way, but we have done a test that is equivalent to the reviewer's suggestion. Since the created rules are all linked to the source mechanisms that were used to generate them, we are able to perform a prediction for an enzyme in the database while removing from the rule set, any rules that contain information only seen in that enzyme. This makes sure that for that calculation, no information from that enzyme is used for generating the output.

The quality of the predictions suffers, as explained in the manuscript, when the test is performed in this way, and that is the reason we identify the rule coverage as the main target for future improvements.

Alternatively, the authors can take a recently elucidated enzyme mechanism not present in the M-CSA and use that to test the algorithm.

As explained before, after taking reviewers' suggestions into consideration we have performed such tests, which are described in the new version of the manuscript.

In the test set, the majority of enzymes have "unique rules". First, what are these unique rules? Second, how often do enzymes exhibit unique rules across the MCSA? These information are not included but would be essential in showing how widely applicable this prediction algorithm would be.

Out of 3668 single-step rules (rules who are seen in its entirety in one or more catalytic steps, as opposed to mixed-step rules which are composed by mixing information from different steps together) 2964 (80%) are unique rules i.e., rules that are seen in only one active site and cannot be reconstructed using data from other enzymes. Among the 691 mechanisms for which we could

extract rules, 521 (75%) mechanisms have unique rules. These numbers mirror the results of our initial validation that has shown that when removing rules from the mechanism from the dataset, we could only get an exact prediction for 25% of the enzymes.

As we already discussed in the first version of the manuscript, we recognise this number is low and one of our main priorities for future work is to improve the coverage of the rules by changing the way they are defined (to make them more generic), or to possibly extend the number of rules by looking at other sources of data other than the M-CSA.

Rules that are seen in only one mechanism are shown among all the rules in the rules page: <https://www.ebi.ac.uk/thornton-srv/m-csa/rules/>, and can be identified by the label: "At least one of the arrows unique to a single entry".

With respect to judging the applicability of EzMechanism, this is a useful number, but it does not mean necessarily that the software would only be helpful for an average of 25% of the new studies:

- Some of the unique rules are similar to other rules which means that a slightly different mechanistic path can probably still be found even if not the exact one shown in the database.
- When looking for enzymes to perform the additional tests (on enzymes that are not in the database) we found that many enzymes being studied were not completely new (several peptidases, lipases, and glycosidases, for example). Here, EzMechanism can find the already known mechanisms, but also find new paths not considered before. An example discussed in the new version of the manuscript is that of COVID protease, where a potential mechanism involving the Ser present in the substrate is suggested.
- For enzymes where EzMechanism cannot find the whole path, it is still possible that it can find some steps in the reactants or products side. Additionally, it is useful for the researchers to know that a given mechanism has not been previously described in the literature (or at least annotated in the M-CSA).

We have clarified the discussion of the validation in the new version of the manuscript to explain that the point of the stricter validation test was to see if EzMechanism can find the exact same mechanism, as annotated in the database, when rules from that enzyme are not included in the calculation.

Line 271: What is the time-complexity of this prediction algorithm as the number of explored configurations increased?

The time-complexity of the algorithm is roughly linear with the increase of configurations explored. This is because, the slowest part of the code, the matching of the rules with each configuration, has typically the same cost regardless of the number of configurations already explored. There are other considerations that might cause the time-complexity to deviate from this linear relationship:

- The matching of the rules to the initial configurations is slower than to later configurations due to the caching of the results, which means that, initially, the time-complexity is less than linear.
- As the number of configurations to explore increases, another part of the algorithm, the one dedicated to find the smallest path (in terms of the prioritisation function) in the configurations graph, may become more relevant for the overall cost of the calculation. This only happens when the number of computed configurations goes past about 100 000. The time complexity of this computation is $O(V^2)$, where v is the number of edges. This is the time complexity of Dijkstra's algorithm.

Other than the number of explored configurations, an important element that greatly affects the speed of the calculation is the average number of matching rules and the number of new configurations generated for an active site. This might be difficult to assess beforehand and so it is difficult to come up with a formula, but the general principle is that bigger active sites and/or active sites that have many acidic/basic groups will lead to the generation of many configurations, due to all the possible combinations of protonation states.

Other feedback:

The prioritization is based on distances of atoms and number of steps and authors noted the challenge of having a huge number of possible steps. Although this opens up more possibilities, would it be possible to use chemical heuristics to simplify the search if users chose to do so? For example, presence of certain cofactors as they would have a set of known mechanistic steps in the database that is the most probable, due to their chemical properties.

This a good idea and one that we might consider in the future. Since some catalytic steps (rules) are seen happening in sequence in more than one enzyme, we could use this chain of rules as a unit of some sort to progress the reaction further in one go.

User friendliness and scripting interface. It seems like prediction needs to be run on an online server and users input molecules by drawing 2D schemes. In future renditions it might be useful to have a python scripting API so that users can run it in high throughput, but this depends on target users and use cases.

We agree, at this point we would like to see how people use the current version of the software before developing further. Indeed, our first approach to the validation of the software was completely automated but we realised that, among other challenges, the selection of the model was difficult to automate, namely the decisions regarding which PDB to use and the selection of substrate in the active site. We might return to this problem in the future.

Reviewer #3:

Remarks to the Author:

The authors describe the generation of 'chemical' rules deduced from known enzyme mechanisms and their application in an automated tool using 3D data of an active site to propose novel and/or alternative mechanisms for enzymes. The manuscript is overall well written and understandable to enzymologists not only to chemists. The tool is of broader interest to the world of enzymology and biocatalysis and may aid the understanding and development of enzymes.

We thank the reviewer for the encouraging comments.

However, I have a few comments:

Major points:

I do understand the important role of M-CSA for this publication, however how this is managed leaves me puzzled that it is partly in the introduction, even explained there, later in the methods. Either reduce this part and cite the reference or use this publication as a kind of update of the 2018 NAR paper and make M-CSA more visible in title and abstract.

We followed the suggestion of the reviewer and simplified the description of the M-CSA in the introduction to avoid repetition.

The part starting from line 53 should be shortened to a single paragraph, it appears partly redundant to the discussion.

We simplified this section to remove redundancy with other parts of the text (from ~500 to ~200 words)

Is there a test case for an enzyme of a yet unknown mechanism available to prove the capacities for hypothesis generation?

As mentioned above we have performed tests on 6 new enzymes that are not present in the database. The results are discussed in the "Validation using Enzymes not annotated in M-CSA" section.

I think it would be good to have this tool available independent of a M-CSA curator account prior to publication.

As explained previously in this document, after taking into account this and similar suggestions, we have built a new web user interface accessible to everyone (after registration) and have written a documentation page to explain its use, available at <https://wwwdev.ebi.ac.uk/thornton-srv/m-csa/EzMechanism/documentation>.

Minor points:

Line 24-25: catalytic macromolecules are not abundant in genomes, genes (putatively) encoding for enzymes are.

We clarified this point in the manuscript:

"These catalytic macromolecules are abundant in all cells (representing 22% of the proteins coded in the human genome and 40% in *E. coli*, for example¹) and are widely studied."

Decision Letter, first revision:

Dear António,

Thank you for submitting your revised manuscript "EzMechanism: An Automated Tool to Propose Catalytic Mechanisms of Enzyme Reactions" (NMETH-A50326B). It has now been seen by the original referees and their comments are below. The reviewers find that the paper has improved in revision, and therefore we'll be happy in principle to publish it in Nature Methods, pending minor revisions to satisfy the referees' final requests and to comply with our editorial and formatting guidelines.

Nature Methods offers a transparent peer review option for new original research manuscripts submitted from 17th February 2021. We encourage increased transparency in peer review by publishing the reviewer comments, author rebuttal letters and editorial decision letters if the authors agree. Such peer review material is made available as a supplementary peer review file. Please state in the cover letter 'I wish to participate in transparent peer review' if you want to opt in, or 'I do not wish to participate in transparent peer review' if you don't. Failure to state your preference will result in delays in accepting your manuscript for publication.

ORCID

Sincerely,
Arunima

Arunima Singh, Ph.D.
Senior Editor
Nature Methods

Reviewer #1 (Remarks to the Author):

In this revised manuscript the authors have presented an improved version of software to predict possible reaction mechanisms from the structure of an enzyme based on rules developed in training their algorithm to define rules based on known enzymes. The the proof of the utility of this approach will not come until users have tested the system after isolating new enzymes and comparing the actual reaction mechanism to predictions. So the question now is whether the somewhat preliminary release of the software and this manuscript fits in with the goals of Nature Methods. An a journal that places a high expectation of novelty, a more complete presentation of the method and its utility years from now would no longer fit the standard. The improvements afforded in the revised version have addressed most of the major reservations and the software appears to be sufficiently refined to turn it over the a wider audience to use and evaluate. A major challenge will be when available ground state structures do not provide knowledge of substantial changes in structure after substrate binding. Presumably, the software will be continually refined with new input. I the final analysis, it seems that the time is right for that to happen.

Reviewer #3 (Remarks to the Author):

As mentioned in the first review, the authors derive rules from known enzyme mechanisms and apply them in a tool that uses 3D data of an active site to propose enzyme mechanisms. The manuscript is well written and understandable to a broad audience with a biochemistry background. The tool is of general interest to the world of enzymology and biocatalysis and can promote the understanding and development of enzymes.

In the revision, the structure of the manuscript has been changed and the separation of introduction and results/discussion is more clearly defined. In particular, the focus is more on EZMechanisms than on

M-CSA. I really like that enzymes not annotated in EZMechanisms are now discussed and that accessibility for users is much easier.

Author Rebuttal, first revision:

Dear Editor,

We would like to thank you and the reviewers for taking the time to evaluate our manuscript a second time and for your positive decision on publication. We now resubmit the manuscript, after taking into account all the editorial suggestions. Since the reviewers did not raise any further concerns in this round of reviews, there were no scientific changes to the manuscript.

[Redacted]

We are at your disposal for any further clarifications.
Thank you once more for your time and help.
António Ribeiro,
on behalf of all co-authors.

Final Decision Letter:

Dear António,

I am pleased to inform you that your Article, "EzMechanism: An Automated Tool to Propose Catalytic Mechanisms of Enzyme Reactions", has now been accepted for publication in Nature Methods. Your paper is tentatively scheduled for publication in our October print issue, and will be published online prior to that. The received and accepted dates will be September 5, 2022 and August 15, 2023. This note is intended to let you know what to expect from us over the next month or so, and to let you know where to address any further questions.

Over the next few weeks, your paper will be copyedited to ensure that it conforms to Nature Methods style. Once your paper is typeset, you will receive an email with a link to choose the appropriate

publishing options for your paper and our Author Services team will be in touch regarding any additional information that may be required.

You will receive a link to your electronic proof via email with a request to make any corrections within 48 hours. If, when you receive your proof, you cannot meet this deadline, please inform us at rjsproduction@springernature.com immediately.

Please note that *Nature Methods* is a Transformative Journal (TJ). Authors may publish their research with us through the traditional subscription access route or make their paper immediately open access through payment of an article-processing charge (APC). Authors will not be required to make a final decision about access to their article until it has been accepted. [Find out more about Transformative Journals](https://www.springernature.com/gp/open-research/transformative-journals)

Your paper will now be copyedited to ensure that it conforms to Nature Methods style. Once proofs are generated, they will be sent to you electronically and you will be asked to send a corrected version within 24 hours. It is extremely important that you let us know now whether you will be difficult to contact over the next month. If this is the case, we ask that you send us the contact information (email, phone and fax) of someone who will be able to check the proofs and deal with any last-minute problems.

If, when you receive your proof, you cannot meet the deadline, please inform us at rjsproduction@springernature.com immediately.

Once your manuscript is typeset and you have completed the appropriate grant of rights, you will receive a link to your electronic proof via email with a request to make any corrections within 48 hours. If, when you receive your proof, you cannot meet this deadline, please inform us at rjsproduction@springernature.com immediately.

Once your paper has been scheduled for online publication, the Nature press office will be in touch to confirm the details.

Once your paper has been scheduled for online publication, the Nature press office will be in touch to confirm the details.

Content is published online weekly on Mondays and Thursdays, and the embargo is set at 16:00 London time (GMT)/11:00 am US Eastern time (EST) on the day of publication. If you need to know the exact publication date or when the news embargo will be lifted, please contact our press office after you have submitted your proof corrections. Now is the time to inform your Public Relations or Press Office about your paper, as they might be interested in promoting its publication. This will allow them time to prepare an accurate and satisfactory press release. Include your manuscript tracking number NMETH-A50326C and the name of the journal, which they will need when they contact our office.

About one week before your paper is published online, we shall be distributing a press release to news organizations worldwide, which may include details of your work. We are happy for your institution or funding agency to prepare its own press release, but it must mention the embargo date and Nature Methods. Our Press Office will contact you closer to the time of publication, but if you or your Press Office have any inquiries in the meantime, please contact press@nature.com.

Nature Portfolio journals [encourage authors to share their step-by-step experimental protocols](https://www.nature.com/nature-research/editorial-policies/reporting-standards#protocols) on a protocol sharing platform of their choice. Nature Portfolio 's Protocol Exchange is a free-to-use and open resource for protocols; protocols deposited in Protocol Exchange are citable and can be linked from the published article. More details can found at www.nature.com/protocolexchange/about.

Best regards,
Arunima

Arunima Singh, Ph.D.
Senior Editor
Nature Methods